# Deficiencies in communication between clinical microbiological laboratories and physicians may impair the diagnosis of Lyme borreliosis: A study of the use and application of serology in three neighbouring counties in Sweden

Marcus Johansson[1,2]*, Henrik Hillerdal[3,4], Matilda Ljungqvist Lövmar[5,6], Lena Serrander[7,8], Anna J. Henningsson[1,2], Ivar Tjernberg[9,10]*

1 Department of Biomedical and Clinical Sciences, Division of Inflammation and Infection, Linköping University, Linköping, Sweden, 2 Department of Laboratory Medicine, Division of Clinical Microbiology, National Reference Laboratory for Borrelia and Other Tick-Borne Bacteria, Jönköping, Region Jönköping County, Sweden, 3 Department of Pediatrics, Jönköping, Region Jönköping County, Sweden, 4 Department of Biomedical and Clinical Sciences, Division of Inflammation and Infection, Linköping University, Linköping, Sweden, 5 Department of Biomedical and Clinical Sciences, Linköping University, Linköping, Sweden, 6 Clinical Department of Clinical Microbiology, Linköping, Region Östergötland, Sweden, 7 Department of Biomedical and Clinical Sciences, Linköping University, Linköping, Sweden, 8 Division of Clinical Microbiology, Linköping University Hospital, Linköping, Sweden, 9 Department of Clinical Chemistry and Transfusion Medicine, Kalmar, Region Kalmar County, Sweden, 10 Department of Biomedical and Clinical Sciences, Division of Inflammation and Infection, Linköping University, Linköping, Sweden

* marcus.johansson@rjl.se (MJ); ivar.tjernberg@liu.se (IT)

## Abstract

### Purpose

The diagnosis of Lyme borreliosis (LB) can be challenging. The aim of this study was to investigate, describe and compare the actual use, application and documentation of LB serology in three neighbouring LB-endemic counties in Sweden. As part of this, we intended to study the concordance between laboratory reports and physicians' assessments regarding LB.

### Methods

Three hundred patients sampled for LB serology in the counties of Jönköping, Kalmar and Östergötland, in 2016 were randomly selected for this study. Data was collected from the laboratory information technology systems of the departments of Clinical Microbiology in the three counties and from medical records.

### Results

Suspected Lyme neuroborreliosis (LNB) was the most common indication for LB serology, and was found in a total of 188/300 (63%) patients: 75/100 in Jönköping,

**Data availability statement:** Data cannot be shared because of restrictions in place that prevent public sharing of original and raw data, according to the regional ethical permission for the study. Data contain potentially identifying and sensitive patient information, including medical history and medication. In order to prevent unpermitted access to the data, public sharing is not allowed. Data sets are available from Linköping University (contact via registra-tor@liu.se) upon request for researchers who meet the criteria for access to confidential data.

**Funding:** This study was supported by university grants to Region Kalmar County (no. 2798) https://www.researchweb.org/is/regionostergot-land (IT) and by the Medical Research Council of South East Sweden (FORSS) (no. 757471), https://www.researchweb.org/is/forss (MJ) The funders had no role in study design, data collection and analysis, decision to publish, or preparation of the manuscript.

**Competing interests:** I.T reports previous participation in an advisory board, personal fees and an ongoing collaboration without personal compensation with Pfizer Inc. outside the submitted work. A.H and M.J reports an ongoing research collaborative agreement without personal compensation with Pfizer Inc. outside the submitted work. This does not alter our adherence to PLOS ONE policies on sharing data and materials.

66/100 in Östergötland and 47/100 in Kalmar. Cerebrospinal fluid examination was performed on a minority of patients in whom LNB was suspected, 34/188 (18%). LB serology was performed on sera from 15 patients with suspected erythema migrans. Sufficient information to enable an assessment of concordance between laboratory reports and medical records was available for 158/300 (53%) patients, while 94/158 (59%) were considered to have concordant records.

## Conclusions

LB serology is frequently performed on questionable indications contrary to guidelines, which limits the value and potential of the analysis. Notably, the use appears to be different in three neighbouring counties that follow the same national guidelines. Although new diagnostic technologies, may improve laboratory diagnostics in the future, there is still a need for interventions to enable a more rational use of LB serology.

## Introduction

Lyme borreliosis (LB) is a tick-borne infectious disease caused by spirochetes of the *Borrelia burgdorferi* sensu lato complex (Bbsl) [1]. The causative agent of LB is transmitted to humans by *Ixodes spp*: in Europe *I. ricinus*, in Asia *I. persulcatus*, in North America *I. scapularis* and *I. pacificus*, in North Africa *I. inopinatus* [2,3]. It is the most common vector-borne disease in temperate North America, Europe and Asia [4–6]. The annual number of clinical cases of LB is estimated to be approximately 230 000 in Western Europe, although this is probably an underestimation [7].

The most common clinical manifestation of LB is erythema migrans (EM), which is present in about 75% of cases and allows a clinical diagnosis [8]. When and if the infecting agent spreads to other tissues and organs, other clinical manifestations of LB may arise, such as Lyme neuroborreliosis (LNB), Lyme arthritis (LA), acrodermatitis chronica athrophicans (ACA) and Lyme carditis [9]. Except for EM, an early localised manifestation of LB characterized by a circular erythema around the tick bite, which does not appear immediately (chemical reaction), but after an incubation period of 4–30 days and enlarges with a greater diameter of 5 cm, diagnosis of LB requires a meticulous synthesis of patient history, clinical examination and laboratory support [10,11].

There are two broad categories of diagnostic tests for LB: direct detection methods, which detect the agent of infection in patient specimens and indirect detection methods, which detect a host response to the infection. *Borrelia* serology is used for the diagnosis of LB in routine clinical practice [12]. Diagnosis of LNB requires simultaneous analysis of *Borrelia*-specific antibodies in serum and cerebrospinal fluid (CSF) [13]. In some cases, when neurological manifestations are secondary to ACA, the cerebrospinal fluid may be negative [14].

Establishing the diagnosis may be complex and challenging before a serological response has developed. However, even in the case of disseminated infection in the joints, the heart, or the central nervous system the diagnosis may be challenging [15].

Serology results should be interpreted in combination with clinical signs and symptoms, including duration of symptoms. However, multiple shortcomings exist. For example, in the early phases of the infection, antibody responses may be absent and antibiotic therapy may abort serologic response or prevent seroconversion [16]. In addition, serum anti-*Borrelia* IgG may persist for decades and therefore, serology cannot be used to monitor disease activity or eradication [17].

A considerable seroprevalence exists in the healthy population. Dong et al. reported a global seroprevalence of 14.5% (95% CI 12.8% to 16.3%) and also confirmed a wide variation in prevalence between regions and countries, with the highest reported prevalence in Central Europe (20.7%), followed by Eastern Asia (15.9%), Western Europe (13.5%) and Eastern Europe (10.4%) [18].

In order to enable adequate interpretations and rational use of serology, it is essential that factors such as clinical features and patient history are taken into consideration [19–21]. To assist physicians in the interpretation process, clinical microbiologists often attach an informative comment to each LB serology laboratory report. In our three neighbouring regions in Sweden (Jönköping, Kalmar and Östergötland), the comments are adapted to the national recommendations, which include information about recommended diagnostic strategies in each manifestation of LB [22]. Also, the instructions regarding indications for testing are adapted to the national recommendations. We consider the information valuable since many physicians of different medical specialities are confronted with suspected LB cases and not all are trained to recognise and diagnose the disease. Numerous scientifically questionable ideas on the clinical presentation, diagnosis and treatment of LB may confuse [10]. To our knowledge, the adherence to this interpretation aid provided by the laboratories has not been evaluated previously. Furthermore, we do not know if LB serology is used similarly in different counties which follow the same national guidelines from the Swedish authorities [22,23]. It should also be noted that serology in the United States and Europe has some differences since the prevalence of different Bbsl species are not the same. Serological tests must take into account these different species of Bbsl [24,25].

The aim of this study was to retrospectively investigate, describe and compare the actual use, application, documentation and interpretation of LB serology in three neighbouring counties in Sweden. As part of this, we intended to study the concordance between laboratory reports and physicians' assessments regarding LB.

## Patients and methods

This study was performed in the counties of Jönköping, Kalmar and Östergötland located in the southeast part of Sweden. The counties cover an area of 31 692 square kilometres. Approximately 1 089 000 people live in the area [26]. This part of Sweden is considered highly endemic for LB. In Kalmar and Jönköping County, a LB-seroprevalence of about 20% in healthy blood donors has been reported [27,28].

LB serology is offered at the Departments of Clinical Microbiology in all three counties. All analyses are performed at one central laboratory in each region. All three laboratories are funded through general taxation and integrated into the public healthcare system.

The assays used for serum samples in Jönköping were Enzygnost Borreliosis IgM and Enzygnost Lyme link VlsE/IgG (Siemens/DADE Behring, Marburg, Germany). The cut-off levels used were 2 U/mL for IgM (adjusted to local background seroreactivity) and 10 U/mL for IgG. Patients with *Borrelia*-specific IgM and/or IgG detected were considered as positive. In Kalmar, all sera were analysed in C6 ELISA (Immunetics, 27 Drydock Ave., Boston, MA, USA). The C6 ELISA uses a conserved synthetic peptide derived from the VlsE protein as an antigen, and both IgM and IgG antibodies are detected in the same assay. VlsE (Variable major protein like sequence Expressed) is an important and specific antigen for diagnosis, as it is not expressed in culture or in ticks, while it is expressed by Bbsl in human infection, inducing an IgG antibody response [29]. Results were considered positive when the signal to cut-off ratio was ≥ 1.10. In Östergötland all sera were

analysed in LIAISON Borrelia IgM and IgG (DiaSorin, Saluggia, Italy). The cut-off levels used were 22 U/mL for IgM and 15 U/mL for IgG. Patients with *Borrelia*-specific IgM and/or IgG detected were considered as positive.

In routine practice, the LB serology requests are sent to the laboratories in digital form. Information regarding clinical manifestation and duration of symptoms should be specified by the requesting physician in order to enable the clinical microbiologists to make an assessment. However, it is technically possible to send a request without providing the information. When the analysis is completed, a report including results and written recommendations is sent back to the requesting physician in digital form. Consequently the laboratory reports not only consist of a measurement of antibody reactivity but also of conclusions drawn by a clinical microbiologist.

We identified all reports for LB serology performed in 2016. Reports were identified using laboratory information technology systems from the Departments of Clinical Microbiology.

Out of a total of 10,095 serology reports, 300 were randomly selected for inclusion in the study—specifically, 100 reports from each of the three counties—using the randomisation function in Microsoft Excel. Consequently patients that were sampled more than once during 2016 had a higher probability to be included in the study. Personal identification numbers were then used to match the laboratory reports with the 300 patients' medical records from both primary and secondary health care. Documentation made in medical records within 365 days starting with the sampling day (day 1) was reviewed. Patients < 18 years of age were not eligible.

Data regarding age, sex, clinical symptoms, symptom duration, indications for testing, LB serology results, assessments and recommendations made by clinical microbiologists, assessments made by physicians (diagnoses), antibiotic treatment for LB, treatment (other than antibiotics) and other diagnoses (other than LB) within 365 days from the sampling day were collected from the databases and medical records by three of the authors (M.J., H.H and M.L), all with long experience of LB.

In order to evaluate the concordance between laboratory reports (assessments made by clinical microbiologists) and medical records (assessments made by physicians), the conclusions of assessments of clinical microbiologists and physicians were compared to each other. Notably, the authors did not evaluate the correctness of the assessments previously made by the treating physician. The comparisons were classified into the following categories:

1. Concordant (same conclusions of the assessments in both laboratory reports and medical records, e.g., LB and LB, or not LB and not LB. Recommendations regarding diagnosis from the clinical microbiologist that were followed by the physicians also fell into this category).

2. Discordant (different conclusions in laboratory reports and medical records, e.g., LB and not LB or not LB and LB. For example, if the physician stated that the patient had ACA, but the clinical microbiologist stated that ACA was unlikely since the serology was negative, the patient fell into this category. Recommendations regarding diagnosis from the clinical microbiologist that were not followed by the physicians also fell into this category).

3. Neither concordant nor discordant (the assessments were documented in laboratory reports and medical records, but were insufficient or did not fulfil the criteria above).

In order to achieve an equivalent assessment level, the three authors regularly discussed and evaluated the assessments together. In particular, all assessments that were considered as difficult and uncertain were discussed among the three authors to achieve consensus.

### Statistics

IBM SPSS statistics 27 was used for statistical calculations with chi-square when comparing frequencies across groups, and the Kruskal-Wallis test was used when comparing age. In case of significant differences when comparing three groups, pairwise analysis with a chi-square or Mann-Whitney U-test was performed.

A p-value < 0.05 was accepted as a significant result.

## Ethics

This study was approved by the Swedish ethical review authority, DNR: 2019−00187. Need for consent was waived by the Swedish ethical review authority. Data were accessed for research purposes at 27 February 2019. Authors had access to potentially identifying information during collection of data.

## Results

The laboratory reports and medical records of all included patients were reviewed. No significant differences were found between the three regions regarding demographic data, except for a higher seropositivity rate in Kalmar County compared to Jönköping and Östergötland (p < 0.001). A majority of patients had been investigated in primary health care, with non-significant variations across regions (Table 1).

LNB was the most common indication for LB serology, being observed for 188/300 (63%) patients, followed by LA at 57/300 (19%), EM 15/300 (5%) and ACA 14/300 (5%). In 14/300 (5%) cases, the LB serology was performed at the patient's own request. Seropositivity rates differed between indications. The rate found in LNB indication patients was 46/188 (24%), which was significantly lower than the seropositivity rate in LA, at 30/57 (53%), and ACA at 12/14 (86%) patients (p < 0.001), (Table 2).

Subanalysis revealed that LNB was the most common indication in all three counties, 75/100 patients in Jönköping, 66/100 in Östergötland and 47/100 in Kalmar. The number of patients with LNB indications was significantly higher in Jönköping and Östergötland compared to Kalmar (p < 0.001 and p < 0.01). LA was the second most common indication in all three counties, 33/100 in Kalmar, 16/100 in Östergötland and 8/100 patients in Jönköping. Patients with LA indications were significantly more common in Kalmar compared to Jönköping and Östergötland (p=<0.001 and p < 0.01), (Table 3).

### Duration of symptoms

Information regarding the duration of symptoms was missing in a majority of laboratory request forms, sent to the clinical microbiological laboratories, 162/300. According to medical records, the median duration of symptoms was 50 days at the time of sampling (range: 1 day-20 years).

### Comments on laboratory reports

To assist physicians in the interpretation process, comments were attached to 194/300 (65%) LB serology laboratory reports.

**Table 1. Demographic data.**

|  | Total | Jönköping | Kalmar | Östergötland | p-value |
|---|---|---|---|---|---|
| Population n (%) | 1 089 000 (100) | 369 000 (34) | 248 000 (23) | 472 000 (43) | N/A |
| LB Serology reports n (%) | 10 095 (100) | 3144 (31) | 3220 (32) | 3731 (37) | N/A |
| Sample size n (% of all reports within each region) | 300 (3.0) | 100 (3.2) | 100 (3.1) | 100 (2.7) | N/A |
| Female/male n (% female) | 158/142 (53) | 54/46 (54) | 53/47 (53) | 51/49 (51) | 0.911 |
| Median (mean) age years | 58 (55.5) | 57 (52.9) | 61 (58.0) | 60 (55.7) | 0.131 |
| Range years | 18-98 | 21-84 | 18-88 | 18-98 | – |
| Primary health care/secondary health care n (% primary health care) | 185/115 (62) | 53/47 (53) | 69/31 (69) | 63/37 (63) | 0.063 |
| Seropositivity n (%) | 100 (33) | 23 (23)***a | 51 (51)***b | 26 (26)***c | <0.001 |
| Previous testing for LB serology n (%) | 110 (37) | 24 (24)**d | 44 (44)**e | 42 (42)**f | 0.005 |

Statistical analysis was performed using chi-square testing except for age, for which the Kruskal-Wallis test was used. In the case of significant differences when comparing three groups (right column) pairwise analysis was performed. Seropositivity: *Borrelia*-specific IgM and/or IgG detected in serum. LB: Lyme borreliosis. N/A: not applicable.

* p < 0.05 ** p < 0.01 d-e, d-f *** p < 0.001a-b, b-c.

**Table 2. Indications for Lyme borreliosis serology and seropositivity in 300 patients.**

| Indications according to medical records | n (seropositivity %) |
|---|---|
| LNB | 188 (24) |
| LA | 57 (53) |
| EM | 15 (40) |
| ACA | 14 (86) |
| Patient request | 14 (29) |
| Others<br>Skin (not EM or ACA), Lymphocytoma, Fever, Ocular features, Cardiac features | 25 (28) |
| Data not available | 5 (0) |
| Two indications | 18 (33) |

LNB: Lyme neuroborreliosis LA: Lyme arthritis EM: Erythema migrans ACA: Acrodermatitis chronica atrophicans.

**Table 3. Differences regarding indications for Lyme borreliosis serology in the three included counties.**

| Indications according to medical records | Jönköping (n = 100) | Kalmar (n = 100) | Östergötland (n = 100) | p-value |
|---|---|---|---|---|
| LNB n (seropositivity %) | 75***[a] (20) | 47***[b] (34) | 66**[c] (23) | <0.001 |
| LA n (seropositivity %) | 8***[a] (25) | 33***[b] (73) | 16**[c] (25) | <0.001 |
| EM n (seropositivity %) | 6 (33) | 5 (60) | 4 (25) | 0.810 |
| ACA n (seropositivity %) | 3 (100) | 7 (86) | 4 (75) | 0.390 |
| Patient request n (seropositivity %) | 3 (33) | 1 (100) | 10 (20) | 0.070 |
| Others n (seropositivity %)<br>(Skin (not EM or ACA), Lymphocytoma, Fever, Ocular features, Cardiac features) | 12 (33) | 8 (13) | 5 (40) | 0.199 |
| Two indications n (seropositivity %) | 9 (44)**[d] | 1 (0)**[e] | 8 (25)*[f] | 0.034 |
| Data not available n (seropositivity %) | 2 (0) | 0 (0) | 3 (0) | 0.116 |

Statistical analysis was performed using chi-square testing. In the case of significant differences when comparing three groups (right column) pairwise analysis was performed. Seropositivity: *Borrelia*-specific IgM and/or IgG detected in serum. LNB: Lyme neuroborreliosis LA: Lyme arthritis EM: Erythema migrans ACA: Acrodermatitis chronica atrophicans.

* p < 0.05 [e-f] ** p < 0.01 [b-c, d-e] * ** p < 0.001[a-b].

## Cerebrospinal fluid examination

Cerebrospinal fluid (CSF) examination was performed for 34/188 (18%) patients with LNB suspicion as part of the investigation, within 365 days from the original serum sampling day. Intrathecally produced *Borrelia*-specific antibodies were detected in 1/34 (3%) patients. Antibodies in sera were detected in 13/34 (38%) patients.

## Concordance between laboratory reports and medical records

Sufficient information to enable an assessment of concordance between laboratory reports and medical records was available for 158/300 (53%) patients (Table 4).

Among discordant cases the indications for LB serology were 29 LNB, 2 LA, 2 EM, 1 ACA and 1 lymphocytoma.

## Antibiotic treatment

Antibiotic treatment of LB was initiated in 59/300 (20%) patients (doxycycline in 47 patients, phenoximethylpenicillin in 11 patients and penicillin G in one patient). In 34/59 (58%) cases, notes supporting antibiotic treatment were absent or insufficient in medical records and/or in laboratory reports (Table 5).

**Table 4. Concordance between assessments/recommendations documented in laboratory reports and medical records.**

| | |
|---|---|
| **Concordant n (%)** | **94/158 (59)** |
| Discordant n (%) | 35/158 (22) |
| Neither concordant nor discordant n (%) | 29/158 (18) |
| N/A[1] n (%) | 142/300 (47) |
| Total n (%) | 300 (100) |

[1]Assessments were missing in laboratory reports and/or medical records.

**Table 5. Indications for antibiotic treatment.**

| Antibiotic treatment | Total[1] n = 60 |
|---|---|
| LNB n (%) | 18/60 (30) |
| EM n (%) | 12/60 (20) |
| ACA n (%) | 12/60 (20) |
| LA n (%) | 9/60 (15) |
| Lymphocytoma n (%) | 2/60 (3) |
| Not specified n (%) | 7/60 (12) |
| Lack of supporting documentation in medical records n (%) | 34/60 (57) |

[1]Two indications in one patient.

LNB: Lyme neuroborreliosis EM: Erythema migrans ACA: Acrodermatitis chronica atrophicans. LA: Lyme arthritis.

Antibodies directed against Bbsl were detected in sera from 45/59 (76%) of patients who received antibiotic treatment.

Subanalysis revealed that the most common indication for antibiotic treatment in Kalmar was LA (9/32) unlike Jönköping and Östergötland where LNB (6/14 in both counties, respectively) was the most common.

CSF examination was performed in 4/18 (22%) patients who received antibiotic treatment for LNB. The median duration of symptoms was 30 days (range: 1 day-16 years) at the time of treatment initiation among these 18 patients.

### Other diagnoses

Other diagnoses than LB were taken into consideration by the physicians in 263/300 (88%) patients, and 168 (56%) patients were diagnosed with a disease/disorder other than LB within a year from sampling. Antibiotic treatment of LB had been initiated in 23 (14%) of these 168 patients. Diagnoses in the category of neurological, psychiatric and musculoskeletal diseases/disorders were most common. However, also brain tumours (two), lung cancer and prostate cancer were diagnosed within a year from sampling (Table 6).

### Discussion

In this study, we investigated and described the actual use of LB serology in three demographically and geographically comparable counties in Sweden. As part of this, we evaluated the concordance between laboratory reports and physicians' assessments regarding LB. To our knowledge, this is the first study describing the use of and adherence to LB serology laboratory reports.

Our most important findings were of a partly questionable and irrational use of LB serology impaired by deficient communication and documentation, which may hamper the correct diagnosis of LB.

**Table 6. Disease/disorder categories.**

| Neurological diseases/disorders n (%) | 38/300 (including 12 patients diagnosed with headache) (13%) |
|---|---|
| Rheumatic diseases n (%) | 22/300 (7%) |
| Psychiatric diseases n (%) | 16/300 (5%) |
| Musculoskeletal diseases n (%) | 16/300 (5%) |
| Cerebrovascular insult n (%) | 6/300 (2%) |
| Skin diseases n (%) | 6/300 (2%) |
| Infections (non-tick associated infections) n (%) | 5/300 (2%) |
| Malignancies n (%) | 5/300 (2%) |
| Dementia n (%) | 4/300 (1%) |
| Other (e.g., non-specific disorders) n (%) | 50/300 (17%) |
| Total n (%) | 168/300 (56%) |

In three neighbouring counties in Sweden, we expected the use of LB serology to be comparable and similar. However, significant differences were detected. In Jönköping and Östergötland an overwhelming majority of patients were sampled on suspicion of LNB (75% vs 66%), while in Kalmar the corresponding number was 47%. In contrast, a significantly larger number of patients were sampled because of suspected LA in Kalmar compared to Jönköping and Östergötland.

We consider it unlikely that the observed differences were caused by differences regarding epidemiology or prevalence of different tick species in the counties. They were more likely due to different traditions, knowledge and expectations in using LB serology.

The seropositivity rate was significantly higher in Kalmar compared to Jönköping and Östergötland. It is important to bear in mind that in contrast to the assays used in Jönköping and Östergötland the C6 ELISA used in Kalmar did not discriminate between IgG and IgM. This may have affected the results. However, the C6 assay has been shown to be comparable to the Liaison Borrelia IgG and IgM [30]. In addition, Tjernberg et al. reported a significantly lower seroprevalence in healthy blood donors in C6 ELISA compared to the Virotech *Borrelia burgdorferi* ELISA IgG/IgM test (VT) (Genzyme Virotech, Russelsheim), and the Liaison Borrelia IgM and IgG chemiluminescence immunoassay (Li CLIA) (DiaSorin; Saluggia, Italy) [31]. In Jönköping, the cut-off for IgM was adjusted to local background seroreactivity, which could also account for some of the differences in seropositivity rates between the counties.

Overall, the most common indication for LB serology in serum was LNB, found in 188/300 (63%) patients. A parallel or subsequent cerebrospinal fluid examination was performed on 34/188 (18%) patients. The European Federation of Neurological Societies (EFNS) guidelines stipulate that the following three criteria should be fulfilled for definite LNB, and two of them for possible LNB: (i) neurological symptoms; (ii) cerebrospinal fluid (CSF) pleocytosis (except for neurological forms secondary to ACA, where there is a spread of *Borrelia* from the skin to the underlying nerve fibers); (iii) *Borrelia*-specific antibodies produced intrathecally [13,14].

Accordingly, we find it interesting that 154 patients with suspected LNB were not subjected to CSF examination. However, it is important to bear in mind that LB serology in serum can sometimes be used to exclude LNB in patients with symptoms lasting > 6–8 weeks [32]. Moreover, it is of interest that intrathecally produced antibodies were detected in only one patient with suspected LNB in this highly endemic part of Sweden. However, 18 patients received antibiotic treatment for LNB. Notably, CSF examination was performed in only 4/18 (22%) patients. It was not possible to determine if cases of possible LNB were among these since data regarding cell count in CSF was not collected.

The second most common indication for LB serology was LA, found in 57/300 (19%) patients. This was rather unexpected since the incidence of LA in Europe is considered to be low [33–35]. It therefore seems unlikely that all nine patients that received antibiotic treatment for LA in Kalmar County actually suffered from LA. No patient received antibiotic treatment for LA in Jönköping and Östergötland.

LB serology was performed on sera from 15 patients with suspected EM. EM is considered a clinical diagnosis that is based on the characteristic appearance of the skin lesion in a patient with the appropriate epidemiologic and exposure history. Serology is not recommended due to low sensitivity [36].

Taken together, our findings indicate an improper and dysfunctional use of LB serology. This confirms findings reported in other studies where up to 82% of LB serological tests were ordered in contradiction to current guidelines [37–39].

Clinical microbiologists in our three regions often attach a comment to each LB serology result, especially when antibodies are detected, in order to assist physicians in the assessments of patients. To enable this it is essential that information regarding clinical symptoms has been documented in the laboratory request forms.

However, information regarding the duration of symptoms was missing in a majority, 162/300 (54%), of laboratory request forms sent to the microbiological laboratories which impaired the possibility for the laboratory to assess the results. This may explain why comments to assist physicians were attached to only 194/300 (65%) LB serology laboratory reports.

One aim of this study was to investigate the concordance between laboratory reports and physicians' assessments regarding LB to clarify if recommendations from clinical microbiologists have significance for physicians' assessments. In order to enable an assessment of concordance and compliance with laboratory reports, it was essential that assessments had been documented in both laboratory reports and in medical records.

However, sufficient information was available for only 158/300 patients. It is worth mentioning that in 22% (35/158) of cases, the assessments were discordant which means that the clinical assessment was not compliant to the recommendation documented in the laboratory report. Our findings thus indicate that documentation of assessments is deficient, which may negatively affect the possibility to adequately investigate and treat patients receiving medical care for suspected LB.

Antibiotic treatment directed against Bbsl was initiated in 59/300 (20%) patients. Interestingly, we found that notes in support of antibiotic treatment were absent or did not justify treatment in 34/59 (58%) medical records. Overall, these findings suggest that antibiotic treatment seems to be initiated on questionable indications in patients with suspected LB.

According to this study, patients investigated for LB are often suffering from other diseases and disorders that should not be mistaken for LB. In 263/300 (88%) patients, at least one other diagnosis was taken into consideration and 168/300 (56%) were diagnosed with a disease/disorder other than LB within a year from sampling. Twenty-three of these 168 patients were treated with antibiotics for LB, implying that the LB serology may have been misleading, thus delaying the true diagnosis. In fact, serology is reliable in establishing who has had LB, while it is often unable to recognize whether the infection is still microbiologically active. Studies are currently underway in the United States to develop proteomics for Bbsl, with a highly sensitive tandem mass spectrometer and with accurate algorithm development, which allows the identification of microbiologically active forms of Lyme in urine [40,41].

The use and application of LB serology would benefit from interventions and more extensive communication between clinical microbiologists and physicians. In order to achieve improvements, the clinical microbiologists should as far as possible always attach a well-written and informative comment to each LB serology report. We also propose that information regarding clinical symptoms and their duration and suspected manifestation should be mandatory on LB serology request forms.

However, the number of included patients in this study which we consider as a pilot study is limited and further studies are encouraged, especially in other regions and countries with different epidemiological situations, taking into account various laboratory diagnostic processes.

## Conclusions

LB serology is frequently performed on questionable indications contrary to guidelines, which limits the value and potential of the analysis. Notably, the use appears to be different in three neighbouring counties that follow the same national guidelines. Although new diagnostic technologies, such as ms/ms proteomics and Bbsl Phage PCR, may improve laboratory diagnostics in the future, there is also a need for interventions to enable a more rational use of LB serology.

 

## Acknowledgments

The authors want to thank Bo Rolander for valuable help with statistics.

## Author contributions

**Conceptualization:** Marcus Johansson, Lena Serrander, Ivar Tjernberg.

**Data curation:** Marcus Johansson, Henrik Hillerdal, Matilda Ljungqvist Lövmar.

**Formal analysis:** Marcus Johansson.

**Funding acquisition:** Marcus Johansson, Ivar Tjernberg.

**Investigation:** Marcus Johansson, Henrik Hillerdal, Matilda Ljungqvist Lövmar.

**Methodology:** Marcus Johansson, Ivar Tjernberg.

**Project administration:** Marcus Johansson.

**Resources:** Marcus Johansson, Lena Serrander, Anna J. Henningsson.

**Supervision:** Marcus Johansson, Ivar Tjernberg.

**Writing – original draft:** Marcus Johansson.

**Writing – review & editing:** Marcus Johansson, Henrik Hillerdal, Matilda Ljungqvist Lövmar, Lena Serrander, Anna J. Henningsson, Ivar Tjernberg.

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
