## [Decision Letter · Decision Letter 0]

7 Aug 2025

Dear Dr. Johansson,

Thank you for submitting your manuscript to PLOS ONE. After careful consideration, we feel that it has merit but does not fully meet PLOS ONE’s publication criteria as it currently stands. Therefore, we invite you to submit a revised version of the manuscript that addresses the points raised during the review process.

We look forward to receiving your revised manuscript.

Kind regards,

Bersissa Kumsa, DVM, MSc, PhD

Academic Editor

PLOS ONE

Journal Requirements:

“I have read the journal's policy and the authors of this manuscript have the following competing interests: I.T reports previous participation in an advisory board, personal fees and an ongoing collaboration without personal compensation with Pfizer Inc. outside the submitted work.

A.H and M.J reports an ongoing research collaborative agreement without personal compensation with Pfizer Inc. outside the submitted work.”

3. We note you have included a table to which you do not refer in the text of your manuscript. Please ensure that you refer to Table 5 in your text; if accepted, production will need this reference to link the reader to the Table.

4. In the online submission form you indicate that your data is not available for proprietary reasons and have provided a contact point for accessing this data. Please note that your current contact point is a co-author on this manuscript. According to our Data Policy, the contact point must not be an author on the manuscript and must be an institutional contact, ideally not an individual. Please revise your data statement to a non-author institutional point of contact, such as a data access or ethics committee, and send this to us via return email. Please also include contact information for the third party organization, and please include the full citation of where the data can be found.

Additional Editor Comments:

Dear Authors,

The reviewers have completed their evaluation of your manuscript. I encourage you to revise and resubmit your work, ensuring that all reviewer comments are thoroughly addressed. Please incorporate the feedback carefully and provide a detailed, point-by-point response that clearly outlines every change made in response to the reviewers’ suggestions.

In addition, kindly correct all typographical and grammatical errors, and ensure that the manuscript is prepared in full compliance with the journal’s formatting and submission guidelines.

We look forward to receiving your revised submission.

Reviewer's Responses to Questions

**Comments to the Author**

1. Is the manuscript technically sound, and do the data support the conclusions?

Reviewer #1: Partly

Reviewer #2: Yes

Reviewer #3: Yes

2. Has the statistical analysis been performed appropriately and rigorously?

Reviewer #1: No

Reviewer #2: Yes

Reviewer #3: Yes

3. Have the authors made all data underlying the findings in their manuscript fully available?

Reviewer #1: No

Reviewer #2: Yes

Reviewer #3: No

4. Is the manuscript presented in an intelligible fashion and written in standard English?

Reviewer #1: Yes

Reviewer #2: Yes

Reviewer #3: Yes

Reviewer #1: Dear editor in chief

While the topic of the article is interesting, the study faces limitations due to the constraints in the techniques employed and the study is conducted at a basic and preliminary level. These limitations may affect the reliability and clinical applicability of the findings. To enhance the potential for publication, it is essential for the authors to address these methodological shortcomings and consider incorporating more robust and validated techniques in their research. This would not only strengthen their conclusions but also increase the overall impact of their work in the field. therefore, it seems that it is not appropriate for publication.

Reviewer #2: The work is interesting and done in a careful way. There are some changes or clarifications to be made:

Borrelia burgdorferi sensu lato abbreviate: Bbsl

In the introduction it is written: The causative agent of LB is transmitted to humans by Ixodes ricinus ticks. This should be corrected to: "Ixodes spp: in Europe Ixodes ricinus, in Asia I. persulcatus, in North America I. scapularis and I. pacificus, in North Africa I. inopinatus".

The most common clinical manifestation of LB is erythema migrans (EM), a clinical diagnosis. Replace with "The most common clinical manifestation of LB is erythema migrans (EM), which is present in 75% of cases and allows a clinical diagnosis.

In the sentence: Except for EM, an early localized manifestation of LB, add "characterized by a circular erythema around the tick bite, which does not appear immediately (chemical reaction), but after an incubation period of 4-30 days and enlarges with a greater diameter of 5 cm. In this early localized phase (stage I) antibodies are often still negative."

The Sentence: Borrelia serology is the cornerstone for the diagnosis of LB in routine clinical practice, I would remove "the cornerstone".

After "Diagnosis of LNB requires simultaneous analysis of Borrelia-specific antibodies in serum and cerebrospinal fluid (CSF), I would add the sentence: "In some cases, when neurological manifestations are secondary to ACA, the cerebrospinal fluid may be negative (Ogrink K, Maraspin V. Nervous System Involvement in Lyme Borreliosis. The Open Dermatology Journal, 2016, 10, (Suppl 1: M5) 44-54. DOI: 10.2174/1874372201610010044)"

After the sentence: Furthermore, we do not know if LB serology is used similarly in different counties which follow the same national guidelines from the Swedish authorities [17, 18]. Add: "It should also be noted that serology in the United States and Europe has some differences. In fact, in North America Borrelia burgdorferi sensu stricto (Bbss) is clearly prevalent, while in Europe Lyme Borreliae, which infect humans, are more numerous and serological tests must take into account these different species of Bbsl. Currently, 24 Borreliae belonging to the Lyme Group have been identified (Trevisan G, Cinco M, Trevisini S, di Meo N, Chersi K, Ruscio M, Forgione P and Bonin S. Borreliae Part 1: Borrelia Lyme Group and Echidna‐Reptile Group. Biology 2021, 10, 1036. https://doi.org/10.3390/biology10101036), and the last isolated on the coasts of California and Borrelia maritima (Margos G, Fedorova N, Becker NS, Kleinjan JE, Marosevic D, Krebs S, Hui L, Fingerle V, Lane RS. Borrelia maritima sp. nov., a novel species of the Borrelia burgdorferi sensulato complex, occupying a basal position to North American species. Int J Syst Evol Microbiol. 2020 Feb;70(2):849-856. doi: 10.1099/ijsem.0.003833.).

Interesting are the Polish studies about the chimeric tests in ELISA and Western (or Immuno)-Blot (Grąźlewska W, Holec-Gąsior L, Sołowińska K, Chmielewski T, Fiecek B, Contreras M. Epitope Mapping of BmpA and BBK32 Borrelia burgdorferi Sensu Stricto Antigens for the Design of Chimeric Proteins with Potential Diagnostic Value. ACS Infect Dis. 2023 Nov 10;9(11):2160-2172. doi: 10.1021/acsinfecdis.3c00258. Epub 2023 Oct 6.).

PATIENTS AND METHODS.

After the sentence: The C6 ELISA uses a conserved synthetic peptide derived from the VlsE protein as an antigen, and both IgM and IgG antibodies are detected in the same assay [22], add the sentence: VlsE (Variable major protein like sequence Expressed) is an important and specific antigen for diagnosis, as it is not expressed in culture or in ticks, while it is expressed by Bbsl in human infection, inducing an IgG antibody response.

DISCUSSION

After the sentence: (ii) cerebrospinal fluid (CSF) pleocytosis, "(except for neurological forms secondary to ACA, where there is a spread of Borreliae from the skin to the underlying nerve fibers)"

After the sentence: Twenty-three of these 168 patients were treated with antibiotics for LB, implying that the LB serology may have been misleading, thus delaying the true diagnosis. Add: In fact, serology is reliable in establishing who has had LB, while it is often unable to recognize whether the infection is still microbiologically active. Studies are currently underway in the United States to develop proteomics for Bbsl, with a highly sensitive tandem mass spectrometer and with accurate algorithm development, which allows the identification of microbiologically active forms of Lyme in urine (Cornero R, Irfan SS, Cachaco S, Zhou W, Byne A, Howard M, McIntyre H, Birkaya B, Liotta L, Luchini A. Identification of Unambiguous Borrelia Peptides in Human Urine Using Affinity Capture and Mass Spectrometry. Methods Mol Biol. 2024;2742:105-122. doi: 10.1007/978-1-0716-3561-2_9.). Another promising method is Borrelia Phage PCR (Shan J, Jia Y, Mijatovic T. Use of Specific Borrelia Phages as a New Strategy for Improved Diagnostic Tests. Methods Mol Biol. 2024;2742:99-104. doi: 10.1007/978-1-0716-3561-2_8.).

CONCLUSIONS

After the sentence: The analysis is frequently performed on questionable indications contrary to guidelines, which limits the value and potential of the analysis. Add: "New diagnostic technologies, such as ms/ms proteomics and Bbsl Phage PCR, may improve laboratory diagnostics in the future."

The paper can be accepted with these changes.

Reviewer #3: Review of PONE-D-24-53373

Title: Deficiencies in communication between clinical microbiological laboratories and physicians may impair the diagnosis of Lyme borreliosis: a study of the use and application of serology in three neighbouring counties in Sweden

Conclusions by authors: LB serology is often used for indications it is not intended for, thereby giving the analysis poor clinical value. Notably, the use appears to be different in three neighbouring counties that follow the same national guidelines. Our findings indicate a need for interventions to enable a more rational use of LB serology.

The authors have used an interesting approach of examining medical and laboratory records to determine the relevance of serological testing for LB. The assessment I qualitative by physicians experienced in the field. The authors do not use defined case definitions, but accept the clinical judgement, as is. This choice can be discussed , but in the opinion of this reviewer reflects the clinical reality and is suitable for the present study.

Interesting study qualifying the well-known overuse of Borrelia serology with clinical data.

The use of serology, without spinal fluid examination, for suspected LNB is highlighted.

The use of antibiotic treatment based on positive serology is also quite frequent, even when the clinical diagnosis is uncertain.

Some detailed remarks and suggestions for clarification:

Page 3: It is stated that “and antibiotic therapy may abort serologic response or prevent seroconversion”. This is an interesting theory and often postulated as an established fact. In the opinion of the reviewer this has never truly been documented, this would require a randomized study with weeks delayed treatment in one arm. This is of course unrealistic. There are two possible mechanisms, early treatment may of course abort antibody generation, but on the other hand treatment may also release lots of bacterial antigens and enhance antigenic stimulation. Both scenarios may occur, but which of these two tendencies are most frequent? But as the word “may” is included in the sentence the wording should be OK.

Page 5:

“randomly selected using Microsoft Excel” Somewhat non-specific. Perhaps something like: using a random function in Microsoft Excel.

More importantly also indicate clearly if records from general practice were available? How were the “recommendations made by clinical microbiologists” available. Often telephone calls are not documented systematically, if at all?

Page 6.

The category 2. is difficult to grasp. How do you construct “conclusions in laboratory reports” The laboratory only measured antibody reactivity? “e.g. LB and not LB or not LB and LB.” is not easy to read. Negative serology is not necessarily discordant with the clinical assessment.

Suggest to clarify.

Page 7. Concerning the data in the sentence starting with “The rate found in LNB indication patients was……” and table 2.

Rates of seropositivity are quite high. Was this the first available testing in all patients? Patients were not referred because of already having tested positive? In the methods suggest to explain more clearly how patients were recruited (as also suggested above). Even in a high endemic setting this rate of seropositivity is high. Did you check that the “random selection” was representative for the positivity rate in the total data?

**Do you want your identity to be public for this peer review?** For information about this choice, including consent withdrawal, please see our Privacy Policy

Reviewer #1: No

Reviewer #2: **Yes: ** Prof. Giusto Trevisan University of Trieste

Reviewer #3: **Yes: ** Ram B. Dessau

---

## [Author Response · Author response to Decision Letter 1]

23 Sep 2025

Response to reviewers

Dear Dr. Kumsa and esteemed reviewers

We sincerely thank you for reviewing our manuscript titled Deficiencies in communication between clinical microbiological laboratories and physicians may impair the diagnosis of Lyme borreliosis: a study of the use and application of serology in three neighbouring counties in Sweden. We greatly appreciate your feedback. Below, we provide responses to comments. All revisions are highlighted in the updated manuscript.

Journal requirements

Answer: The manuscript has been adapted to PLOS ONE’S style requirements.

2. “Thank you for stating the following in the Competing Interests section

I have read the journal's policy and the authors of this manuscript have the following competing interests: I.T reports previous participation in an advisory board, personal fees and an ongoing collaboration without personal compensation with Pfizer Inc. outside the submitted work.

A.H and M.J reports an ongoing research collaborative agreement without personal compensation with Pfizer Inc. outside the submitted work.”

Please include your updated Competing Interests statement in your cover letter; we will change the online submission form on your behalf”.

Answer: We accept to update the competing interest section to: I.T reports previous participation in an advisory board, personal fees and an ongoing collaboration without personal compensation with Pfizer Inc. outside the submitted work. A.H and M.J reports an ongoing research collaborative agreement without personal compensation with Pfizer Inc. outside the submitted work. This does not alter our adherence to PLOS ONE policies on sharing data and materials.

3. “We note you have included a table to which you do not refer in the text of your manuscript. Please ensure that you refer to Table 5 in your text; if accepted, production will need this reference to link the reader to the Table”.

Answer: The text has been supplemented according to the request.

4. “In the online submission form you indicate that your data is not available for proprietary reasons and have provided a contact point for accessing this data. Please note that your current contact point is a co-author on this manuscript. According to our Data Policy, the contact point must not be an author on the manuscript and must be an institutional contact, ideally not an individual. Please revise your data statement to a non-author institutional point of contact, such as a data access or ethics committee, and send this to us via return email. Please also include contact information for the third party organization, and please include the full citation of where the data can be found”.

Answer: The text has been amended. “There are restrictions in place that prevent public sharing of original and raw data according to the regional ethical permissions for the study. Data contain potentially identifying and sensitive patient information and public sharing is not allowed, please contact Linköping University at registrator@liu.se regarding access to the data”.

Reviewer #1

“While the topic of the article is interesting, the study faces limitations due to the constraints in the techniques employed and the study is conducted at a basic and preliminary level. These limitations may affect the reliability and clinical applicability of the findings. To enhance the potential for publication, it is essential for the authors to address these methodological shortcomings and consider incorporating more robust and validated techniques in their research. This would not only strengthen their conclusions but also increase the overall impact of their work in the field. Therefore, it seems that it is not appropriate for publication”.

Answer: We are well aware that this study has limitations, such as a limited number of included patients and categorizations of assessment that are difficult to validate. However, this study has the character of a pilot study. To our knowledge, a study that examines the actual use and correlation between laboratory results and physicians assessments has never been made regarding Lyme borreliosis and therefore it is a challenge to decide which robust and validated techniques that are more suitable to this study.

Our results indicate that the use of LB serology is irrational, and that interventions would be of benefit. Our regions are probably not unique, and therefore we consider it valuable to encourage further studies in other regions and countries to increase the possibilities for a more rational diagnosis of LB by publishing the results.

Reviewer #2

“The work is interesting and done in a careful way. There are some changes or clarifications to be made:

Borrelia burgdorferi sensu lato abbreviate: Bbsl”

Answer: The text has been amended according to reviewers request.

“In the introduction it is written: The causative agent of LB is transmitted to humans by Ixodes ricinus ticks. This should be corrected to: "Ixodes spp: in Europe Ixodes ricinus, in Asia I. persulcatus, in North America I. scapularis and I. pacificus, in North Africa I. inopinatus".

Answer: The text has been amended according to reviewers request.

“The most common clinical manifestation of LB is erythema migrans (EM), a clinical diagnosis. Replace with "The most common clinical manifestation of LB is erythema migrans (EM), which is present in 75% of cases and allows a clinical diagnosis”.

Answer: The text has been amended according to reviewers request.

“In the sentence: Except for EM, an early localized manifestation of LB, add "characterized by a circular erythema around the tick bite, which does not appear immediately (chemical reaction), but after an incubation period of 4-30 days and enlarges with a greater diameter of 5 cm. In this early localized phase (stage I) antibodies are often still negative."

Answer: The text has been amended according to reviewers request.

“The Sentence: Borrelia serology is the cornerstone for the diagnosis of LB in routine clinical practice, I would remove "the cornerstone".

Answer: The text has been amended according to reviewers request.

“After Diagnosis of LNB requires simultaneous analysis of Borrelia-specific antibodies in serum and cerebrospinal fluid (CSF), I would add the sentence: "In some cases, when neurological manifestations are secondary to ACA, the cerebrospinal fluid may be negative”.

Answer: The text has been amended according to reviewers request.

“After the sentence: Furthermore, we do not know if LB serology is used similarly in different counties which follow the same national guidelines from the Swedish authorities [17, 18]. Add: "It should also be noted that serology in the United States and Europe has some differences. In fact, in North America Borrelia burgdorferi sensu stricto (Bbss) is clearly prevalent, while in Europe Lyme Borreliae, which infect humans, are more numerous and serological tests must take into account these different species of Bbsl. Currently, 24 Borreliae belonging to the Lyme Group have been identified”.

Answer: The text has been adapted to reviewers request.

“After the sentence: The C6 ELISA uses a conserved synthetic peptide derived from the VlsE protein as an antigen, and both IgM and IgG antibodies are detected in the same assay [22], add the sentence: VlsE (Variable major protein like sequence Expressed) is an important and specific antigen for diagnosis, as it is not expressed in culture or in ticks, while it is expressed by Bbsl in human infection, inducing an IgG antibody response”.

Answer: The text has been amended according to reviewers request.

“After the sentence: (ii) cerebrospinal fluid (CSF) pleocytosis, "(except for neurological forms secondary to ACA, where there is a spread of Borreliae from the skin to the underlying nerve fibers)"

Answer: The text has been amended according to reviewers request.

“After the sentence: Twenty-three of these 168 patients were treated with antibiotics for LB, implying that the LB serology may have been misleading, thus delaying the true diagnosis. Add: In fact, serology is reliable in establishing who has had LB, while it is often unable to recognize whether the infection is still microbiologically active. Studies are currently underway in the United States to develop proteomics for Bbsl, with a highly sensitive tandem mass spectrometer and with accurate algorithm development, which allows the identification of microbiologically active forms of Lyme in urine”

Answer: The text has been amended according to reviewers request.

“After the sentence: The analysis is frequently performed on questionable indications contrary to guidelines, which limits the value and potential of the analysis. Add: "New diagnostic technologies, such as ms/ms proteomics and Bbsl Phage PCR, may improve laboratory diagnostics in the future."

Answer: The text has been amended according to reviewers request.

Reviewer #3

“It is stated that “and antibiotic therapy may abort serologic response or prevent seroconversion”. This is an interesting theory and often postulated as an established fact. In the opinion of the reviewer this has never truly been documented, this would require a randomized study with weeks delayed treatment in one arm. This is of course unrealistic. There are two possible mechanisms, early treatment may of course abort antibody generation, but on the other hand treatment may also release lots of bacterial antigens and enhance antigenic stimulation. Both scenarios may occur, but which of these two tendencies are most frequent? But as the word “may” is included in the sentence the wording should be OK”.

Answer: Thank you for valuable input!

“randomly selected using Microsoft Excel” Somewhat non-specific. Perhaps something like: using a random function in Microsoft Excel”.

Answer: The text has been amended according to reviewers request.

“More importantly also indicate clearly if records from general practice were available? How were the “recommendations made by clinical microbiologists” available. Often telephone calls are not documented systematically, if at all”?

Answer: Records from general practice/primary health care were fully available. The recommendations made by clinical microbiologists were documented in the laboratory reports that were sent to the physicians in digital form.

The patients and methods section has been supplemented.

“The category 2. is difficult to grasp. How do you construct “conclusions in laboratory reports” The laboratory only measured antibody reactivity? “e.g. LB and not LB or not LB and LB.” is not easy to read. Negative serology is not necessarily discordant with the clinical assessment”.

Answer: The laboratory reports not only consist of a measurement of antibody reactivity but also of conclusions drawn by a clinical microbiologist. In order to make these conclusions, the clinical microbiologist has to take antibody reactivity and clinical information provided by the physician in the laboratory request forms into consideration. For example, if the physician stated that the patient had ACA, but the clinical microbiologist stated that ACA is unlikely since the serology is negative, the patient fell into this category.

The patients and methods section has been supplemented.

“Concerning the data in the sentence starting with “The rate found in LNB indication patients was……” and table 2. Rates of seropositivity are quite high. Was this the first available testing in all patients? Patients were not referred because of already having tested positive? In the methods suggest to explain more clearly how patients were recruited (as also suggested above). Even in a high endemic setting this rate of seropositivity is high. Did you check that the “random selection” was representative for the positivity rate in the total data?”

Answer: All 10095 samples from 2016 had an equal chance to be included in the study. Consequently patients that were sampled more than once during 2016 had a higher probability to be included. This selection model was chosen so the study would reflect the clinical reality. We found pre-existing test results for LB serology in 110/300 cases in the laboratory data system.

The seropositivity rates are high especially in Kalmar County and not necessarily representative for the positivity rate in the total data. The study was not designed to estimate the seropositivity rate and we consider the rates as a secondary finding.

The patients and methods section as well as table 1 have been supplemented.

Sincerely,

Marcus Johansson

On behalf of all authors.

---

## [Decision Letter · Decision Letter 1]

9 Dec 2025

Deficiencies in communication between clinical microbiological laboratories and physicians may impair the diagnosis of Lyme borreliosis: a study of the use and application of serology in three neighbouring counties in Sweden.

PONE-D-24-53373R1

Dear Dr. Johansson,

We’re pleased to inform you that your manuscript has been judged scientifically suitable for publication and will be formally accepted for publication once it meets all outstanding technical requirements.

Kind regards,

Pablo Colunga-Salas

Academic Editor

PLOS One

Additional Editor Comments (optional):

Dear Authors,

Based on the comments from two of the three reviewers, as well as the changes made to the manuscript, I appreciate your commitment and contribution to the study of Lyme borreliosis. Therefore, I believe your manuscript is ready for acceptance and to continue with the editorial process.

Reviewers' comments:

Reviewer's Responses to Questions

**Comments to the Author**

Reviewer #1: (No Response)

Reviewer #2: All comments have been addressed

Reviewer #3: All comments have been addressed

2. Is the manuscript technically sound, and do the data support the conclusions?

Reviewer #1: (No Response)

Reviewer #2: Yes

Reviewer #3: Yes

3. Has the statistical analysis been performed appropriately and rigorously?

Reviewer #1: (No Response)

Reviewer #2: Yes

Reviewer #3: Yes

4. Have the authors made all data underlying the findings in their manuscript fully available?

Reviewer #1: (No Response)

Reviewer #2: Yes

Reviewer #3: Yes

5. Is the manuscript presented in an intelligible fashion and written in standard English?

Reviewer #1: (No Response)

Reviewer #2: Yes

Reviewer #3: Yes

Reviewer #1: (No Response)

Reviewer #2: The requested changes have been made correctly. The difference between Borrelia burgdorferi sensu laro and Borrelia burgdorferi sensu stricto has been clarified, and the spread of Lyme borreliosis to other areas of North America and Europe has been clarified. The description of Erythema migrans and its evolution is now thorough, and the neurological and cardiac clinical manifestations are better defined.

The description of the tests has been improved, also with an eye to the future, with new techniques currently under development.

The work can be accepted.

Reviewer #3: No further comments.

However the sentence: Although new diagnostic technologies, such as ms/ms proteomics and Bbsl Phage

PCR, may improve laboratory diagnostics in the future, there is also a need for interventions to enable a more

rational use of LB serology.

Is not really a part of the conclusion - should perhaps appear as a perspective in the discussion. Also these techniques are quite preliminary ideas prestented at conference meetings/publications. Even if borrelia detection is possible clinical sensitivities may be quite low?

**Do you want your identity to be public for this peer review?** For information about this choice, including consent withdrawal, please see our Privacy Policy

Reviewer #1: **Yes: ** Shadi Aghamohammad

Reviewer #2: **Yes: ** Giusto Trevisan, University of Trieste, Italy

Reviewer #3: **Yes: ** Ram B. Dessau

---

## [Editor Report · Acceptance letter]

PONE-D-24-53373R1

PLOS One

Dear Dr. Johansson,

I'm pleased to inform you that your manuscript has been deemed suitable for publication in PLOS One. Congratulations! Your manuscript is now being handed over to our production team.

Kind regards,

on behalf of

Pablo Colunga-Salas

Academic Editor

PLOS One